# CAN CONFIDENCE ESTIMATES DECIDE WHEN CHAIN-OF-THOUGHT IS NECESSARY FOR LLMS?

## ABSTRACT

Chain-of-thought (CoT) prompting has emerged as a common technique for enhancing the reasoning abilities of large language models (LLMs). While extended reasoning can boost accuracy on complex tasks, it is often unnecessary and substantially increases token usage, limiting the practicality of reasoning models in many scenarios. Recent models, such as GPT-OSS and Qwen3, expose controls that enable users to adjust the length of CoT or determine whether it is used at all. Yet, it remains unclear when CoT should be used: on some tasks it improves performance, while on others it provides little benefit or even harms performance. We address this challenge with confidence-gated CoT, where a model invokes reasoning only when confidence in its direct answer is low. To this end, we present the first systematic study of training-free confidence estimation methods for CoT gating. Specifically, we evaluate four training-free confidence estimation methods and compare them to a random baseline and an oracle that always knows when CoT is needed. Through extensive experiments, we show that existing training-free confidence measures can reduce redundant CoT and outperform randomly invoked CoT. However, the utility of individual confidence measures is inconsistent, varying with both the dataset and the model, underscoring the difficulty of deploying confidence-gated CoT in practice. By analysing both strengths and failure modes, our study highlights the potential and limitations of current methods and paves the way toward more reliable adaptive gating of CoT.[1]

## 1 INTRODUCTION

Chain-of-thought (CoT) prompting (Wei et al., 2022; Guo et al., 2025) has become a cornerstone for improving the reasoning capabilities of large language models (LLMs). By encouraging models to generate step-by-step explanations before producing an answer, CoT consistently improves accuracy on tasks requiring multi-step reasoning, such as mathematics, symbolic reasoning, and scientific question answering (Wei et al., 2022; Guo et al., 2025; Qwen Team, 2025). However, extended reasoning is not always beneficial. For many queries, additional reasoning provides limited benefit and sometimes harms accuracy, while substantially increasing token usage and latency (Liu et al., 2024; Sprague et al., 2025). This inefficiency can limit the practicality of reasoning-augmented LLMs where efficiency is important.

Recent models such as GPT-OSS (OpenAI, 2025) and Qwen3 (Qwen Team, 2025) provide a *hybrid thinking mode* that lets users control when and how much reasoning the model produces. However, deciding whether CoT is necessary falls on the user, who must anticipate the difficulty of each query. Adaptive reasoning methods aim to relieve this burden by automatically adjusting reasoning depth. Most past work relies on reinforcement learning or classifiers to predict when CoT helps (Yue et al., 2025; Jiang et al., 2025; Chuang et al., 2025a). These are powerful, but they require additional training. Other work explores training-free indicators such as perplexity (Lu et al., 2025). *Our work generalises this idea under the broader notion of* **confidence-gating** *(Figure 1), where confidence signals are used to decide whether the model should answer directly or switch to CoT.*

Confidence scores give a simple signal of how reliable a model's answer is (Kadavath et al., 2022; Kuhn et al., 2023; Farquhar et al., 2024). They can be verbalised directly by the model (Tian et al.,

---

[1]Our anonymous code is available on https://anonymous.4open.science/r/cgr-DDCE

2023) or derived from its output probabilities Kadavath et al. (2022). They have already been used in model routing (Ramirez et al., 2024; Chuang et al., 2025b), where easy queries are sent to smaller models and harder ones to larger models. This motivates our central questions: *can self-assessed confidence guide LLMs in deciding when to invoke CoT reasoning?*

Our objective is to activate CoT only when necessary, reducing redundant tokens while preserving accuracy. We call this approach confidence-gated CoT. To evaluate this, we benchmark four representative self-assessed confidence methods across diverse reasoning benchmarks within our confidence-gated CoT. We frame this as a gating problem, where each query is routed either to direct answer or to CoT reasoning. To put the results in context we compare against two baselines: the expected performance of random gating and an oracle that always knows when CoT is required. The four approaches we evaluate are: asking the model to state its own certainty (verbalised confidence) (Tian et al., 2023), using the answer's perplexity, asking whether its answer is correct($P$(True)) (Kadavath et al., 2022), and comparing the probabilities of the top two tokens (margin) Ramirez et al. (2024). We measure accuracy and token cost.

Our main findings are:

- Training-free confidence measures can reduce redundant CoT and can consistently outperform random gating, but fall short of the oracle.

- Larger models benefit more from confidence-based gating, but performance varies across tasks.

- In realistic settings, confidence-gated CoT can preserve accuracy while reducing CoT usage by 25–30%, therefore substantially reducing overall token cost.

- Oracle policies highlight a big opportunity for improvement: saving more than 500 tokens on average per query and achieving 4–5% higher accuracy.

We identify both strengths and failure cases with qualitative analysis, underscoring the potential challenges of deploying confidence-gated CoT in practice. Overall, our results provide the first empirical foundation for understanding how self-assessed confidence can support adaptive reasoning. We outline clear directions for developing more reliable strategies to decide when LLMs should "think step-by-step" and when they can answer directly.

Figure 1: Confidence-gated CoT controls if a query is answered directly or with reasoning: high-confidence queries are answered directly, while low-confidence ones trigger reasoning.

## 2 RELATED WORK

### 2.1 ADAPTIVE REASONING

In order to mitigate overthinking, adaptive reasoning aims to enable LLMs to dynamically adjust the depth or length of their reasoning processes based on certain indicators (Yue et al., 2025). Adaptive reasoning methods typically adopt reinforcement learning (RL) frameworks, where carefully-designed reward mechanisms guide LLMs to learn strategies under varying conditions (Jiang et al., 2025; Wang et al., 2025; Luo et al., 2025; Chung et al., 2025). Cheng et al. (2025) propose the Adaptive Cognition Policy Optimisation (ACPO) framework and an online token length budget (TLB) to enable dynamic switches between fast and slow thinking based on the estimated task difficulty. Liu et al. (2025) propose a classification-based method, which leverages features of the token probability distribution, to predict whether CoT will provide gains and switch between direct answers and CoT. The papers mentioned above rely on RL, while there are other works employ training-free estimators. Zhu et al. (2025) use entropy and token probability to decide if CoT is necessary to generate each line of code during code generation. Lu et al. (2025) introduce Certainty-based Adaptive Reasoning (CAR) that uses the perplexity of a direct answer to decide if the model should think for longer. However, there is no research on systematically evaluating which training-free estimator is best suited for adaptive reasoning across diverse tasks.

## 2.2 Model Cascades and Routing

Different from making a specific LLMs adapt to multiple reasoning modes, model cascades and routing dynamically switching between multiple models. Ong et al. (2025) propose to decide when to route based on a win prediction model that estimates the probability of a strong model win over a weak model for a given query. Feng et al. (2025) predict the effect and cost of potential edges in a graph where the task, query, and LLM are modelled as heterogeneous nodes. Ramirez et al. (2024) find that simple confidence measures can effectively route harder queries to stronger models compared to trained routing models. Chuang et al. (2025b) investigates a comprehensive set of self-assessed confidence estimation methods for model routing.

## 2.3 Confidence and Uncertainty Estimation in LLMs

Confidence and uncertainty are two closely related concepts which are often used for gauging the trustworthiness of responses generated by LLMs (Zhu et al., 2025; Chuang et al., 2025b). Lin et al. (2024) give the following differentiation: uncertainty reflects the variability of a model's predictions, while confidence estimates the probability that a specific prediction is correct. Uncertainty is often estimated by sampling multiple responses and measuring their semantic diversity (Kuhn et al., 2023; Farquhar et al., 2024). Semantic entropy clusters equivalent answers and computes entropy over aggregated probabilities, outperforming logit-based baselines such as $P(True)$ (Kuhn et al., 2023; Farquhar et al., 2024) at detecting hallucinations, though it requires multiple samples. To reduce this cost, Kossen et al. (2025) predict semantic entropy directly from model activations. Shifting Attention to Relevance (SAR) reweights token entropy by their importance to the final answer, also relying on multiple sampling to get token importance scores(Duan et al., 2024). As our focus is on single-pass, training-free methods with small overhead, we do not include these sampling-based approaches in our evaluation.

## 3 Confidence-Gated Chain-of-Thought

We propose confidence-gated CoT, where a model selectively triggers reasoning based on its self-assessed confidence. Each query is first answered directly. If the confidence score is low, the model re-runs the query with CoT enabled. We systematically evaluate using four confidence estimation methods: *perplexity*, $P(True)$, *margin sampling*, and *verbalised confidence*.

### 3.1 Problem definition

We study the decision of whether a model should stop after a direct answer or answer with CoT reasoning. For each input $x_i$, the model first generates a direct answer. A direct answer and confidence score $s(x_i; \theta)$ is then derived from a model parametrised by $\theta$. If the score is above the threshold $\tau$, the direct answer is accepted; otherwise, the model answers the question with CoT enabled:

$$gate(x_i; \tau, \theta) = \begin{cases} \text{CoT}(x_i; \theta), & s(x_i; \theta) < \tau \\ \text{DIRECT}(x_i; \theta), & s(x_i; \theta) \geq \tau, \end{cases}$$

This differs from early-exit methods, which require generating partial reasoning before deciding to stop (Yang et al., 2025). In our formulation, reasoning is skipped entirely when the confidence in the direct answer is sufficient. These two approaches are complementary since confidence gating selects when to trigger reasoning and early exiting can still be applied once CoT has been selected.

**Chain-of-Thought:** This mode triggers the model to generate an explicit intermediate reasoning trace before emitting a concise final answer. Specifically, we use the thinking mode of Qwen3 or GPT-OSS, which triggers multi-step reasoning by inserting a special instruction in the prompt (Qwen Team, 2025; OpenAI, 2025).

**Direct:** The model is instructed to output only the final answer without generating intermediate reasoning. To enforce this behaviour, we append a concise instruction such as "*Answer:*" to the prompt, which reliably elicits a short response with no CoT or explanation.

## 3.2 Self-Assessed Confidence

In this study, we limit the scope within self-assessed confidence, where the confidence scores are produced by the model itself or computed based on its outputs without using another predictor. All strategies we study can be generated without sampling answers multiple times and without additional training. These methods have low inference overhead and are broadly applicable.

**Perplexity:** In our study, we view the perplexity of the generated direct answer as a measure of the LLM's confidence in it. Given a direct answer sequence $y = (y_1, \ldots, y_T)$ with $T$ tokens, perplexity is defined as:

$$\text{PPL}(y \mid x_i) = \exp\left( -\frac{1}{T} \sum_{t=1}^{T} \log p(y_t \mid y_{<t}, x_i) \right).$$

A higher perplexity indicates lower confidence in the generated answer.

$P(\textbf{True})$ **(Kadavath et al., 2022):** This approach first generates an answer via direct prompting. Then, we ask the LLM whether the generated answer is (A) True or (B) False in a second forward pass. We then extract the probability of generating the token "A". Full prompt details are found in Appendix D.

**Margin Sampling:** This method measures the difference of the probabilities between the most likely and second most likely predictions produced by the model for a given input. Margin sampling has been used with success for model cascades (Ramirez et al., 2024).

**Verbalised Confidence:** This approach prompts off-the-shelf LLMs to self-evaluate and express its confidence as part of its response (Yang et al., 2024). Following prior work (Yang et al., 2024; Tian et al., 2023), we ask the model to output a confidence score between 0.0 and 1.0 after its answer, which has shown to provide good calibration. Full prompt details are found in Appendix D.

## 3.3 Budgets and Pareto-Optimal Thresholds

We define the CoT budget as the proportion of queries that trigger CoT. This reflects scenarios where only a limited fraction of queries can be allocated to the more costly reasoning mode. To vary this budget, we sweep percentiles of the confidence score distribution, which provides a fixed fraction of queries to be routed to CoT. This allows us to trace accuracy-efficiency trade-offs across different budgets, plotting accuracy against average token cost or CoT usage.

We also explore a practical method for identifying Pareto-optimal thresholds. A threshold is Pareto-optimal if no other setting achieves equal or higher accuracy at lower token cost. The set of such thresholds forms the Pareto front, which traces the best accuracy-cost trade-offs. In practice, we are interested in finding the point in this front with the lowest token cost whose accuracy is within a tolerance $\epsilon$ of the use CoT all the time:

$$\tau^* = \arg\min_{\tau} \text{Tok}(\tau) \quad \text{s.t. } \text{Acc}(\tau) \geq \text{Acc}_{\text{All-CoT}} - \epsilon.$$

To simulate realistic deployment, thresholds are estimated from a calibration set. We sweep percentiles, construct the Pareto front, and select $\tau^*$. The chosen threshold is then applied to the held-out test set. To account for variability in calibration splits, we repeat the procedure multiple times with random fixed-size calibration/test partitions via Monte Carlo cross-validation (Xu & Liang, 2001). We report the mean and standard deviation of accuracy and average tokens per query across runs. This tests if confidence gating can realistically preserve accuracy while reducing cost.

**Online vs Offline Evaluation** We consider both offline and online settings for estimating percentile thresholds. In the offline case, all direct answers and confidence scores are computed first, giving access to the full distribution of confidence scores before any decision is made. This allows thresholds to be set exactly at chosen percentiles. In the online case, we simulate streaming input queries so thresholds must be decided on the fly without access to the overall confidence score distribution. We follow the dynamic percentile method introduced by Ramirez et al. (2024). After each query $t$, the threshold $\tau_t$ is set to the $p$-th percentile of $\{s(x_1), \ldots, s(x_{t-1})\}$. We randomise

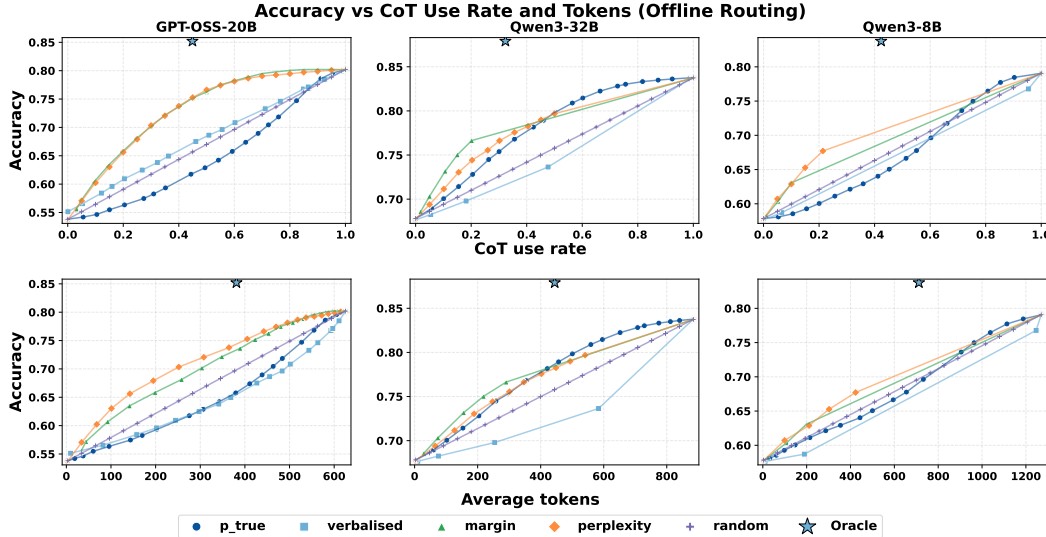

Figure 2: **Offline accuracy–efficiency trade-offs under percentile budgets.** Accuracy vs. CoT usage (top) and vs. average tokens (bottom), aggregated over all datasets for GPT-OSS-20B (medium effort), Qwen3-32B, and Qwen3-8B. Curves show verbalised, perplexity, $P$(True), and margin vs. the random baseline; stars denote the oracle. Full GPT-OSS results for low/medium/high effort are in Appendix E.

dataset order and use a short warm-up phase (the first 20 queries answered directly) to initialise the observations, and report the mean and standard deviation over 10 runs.

## 4 EXPERIMENTAL SETUP

### 4.1 MODELS

Hybrid-reasoning models allow the user to choose between **thinking** and **non-thinking modes** (Qwen Team, 2025). We extend this definition to GPT-OSS, which allows the user to choose between low, medium and high reasoning effort (CoT length) in the prompt (OpenAI, 2025). *We focus on hybrid models such as Qwen3 and GPT-OSS*, which natively support confidence-based gating. In contrast, non-hybrid models without controllable modes require additional instructions or mechanisms to selectively enable reasoning at inference time. GPT-OSS supports three CoT effort settings (*low*/*medium*/*high*) controlled via prompt. Unless specified, the CoT results of GPT-OSS were generated with the *medium* setting. We provide the results of the other effort levels in Appendix E.

### 4.2 DATASETS

The experiments include seven datasets (statistics in Appendix D Table 3) from four reasoning types (Sprague et al., 2025): (1) *commonsense reasoning* including CommonsenseQA (CSQA) (Talmor et al., 2019) and StrategyQA (Geva et al., 2021); (2) *knowledge-based reasoning* using MMLU-redux (Gema et al., 2025); (3) *mathematical and scientific reasoning* on GPQA (Rein et al., 2024) and GSM8k (Cobbe et al., 2021); and (4) *soft reasoning* using LSAT-AGI (Zhong et al., 2024) and MUSR (Sprague et al., 2024). Following (Sprague et al., 2025), these are multiple choice or short answer tasks as CoT is not used as frequently for long-form responses. This wide range of reasoning types allows us to test datasets where reasoning has shown different levels of effectiveness.

### 4.3 BASELINES

**Expected Random Baseline.** For a given CoT usage budget $r \in [0, 1]$, we report the expected accuracy and token cost: $\text{Acc}_r = (1-r)\,\text{Acc}_{\text{Direct}} + r\,\text{Acc}_{\text{CoT}}, \quad \text{Tok}_r = (1-r)\,\text{Tok}_{\text{Direct}} + r\,\text{Tok}_{\text{CoT}}.$

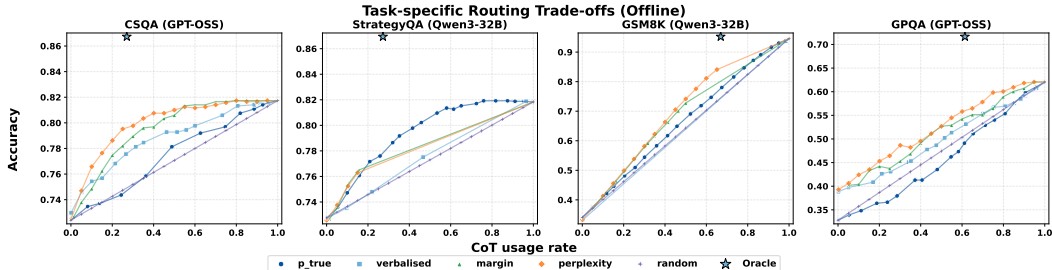

Figure 3: **Task-level accuracy–efficiency trade-offs.** Representative datasets (CSQA, StrategyQA, GSM8K, GPQA) comparing confidence-gating to random and oracle across models.

We compute these analytically rather than by randomly selecting for each point, which provides a fairer and more stable baseline.

**Oracle.**    To assess the ceiling of confidence-based gating, we include an oracle method that triggers CoT whenever the direct answer is incorrect. This setting assumes perfect knowledge of correctness and therefore represents the maximum performance that any confidence signal could achieve. The oracle thus serves as an upper bound on the potential of confidence-guided CoT routing.

## 5    EXPERIMENTAL RESULTS

First, we look at the offline setting, where thresholds are chosen with access to the score distribution over the entire datasets. This provides a clear view of the trade-offs between accuracy and efficiency at different CoT budgets. We then turn to the online setting to see if these trade-offs hold on streaming inputs. Finally, the Pareto-optimal analysis identifies settings that maintain accuracy while lowering token cost.

**CoT Budget–Accuracy Trade-offs.**    We evaluate accuracy–efficiency curves by sweeping percentile budgets as defined in §3.3. At each budget level, we report both accuracy and average token usage. Figure 2 shows aggregate results for GPT-OSS-20B, Qwen3-32B, and Qwen3-8B, comparing confidence-based gating against random selection and the oracle.

For both GPT-OSS-20B and Qwen3-32B, there are confidence methods that achieve clear wins over the random baseline. Specifically, *margin* and *perplexity* consistently outperform random for GPT-OSS-20B, while $P(True)$ is most effective for Qwen3-32B. Using these methods, both models can match the accuracy of always using CoT while invoking it roughly 30–40% less often, showing that confidence can cut token usage effectively at different budgets. For Qwen3-8B, sometimes *margin* and *perplexity* outperform random at low percentiles and $P(True)$ at higher ones, but no method consistently beats random across all budgets. The oracle highlights that large efficiency improvements are possible, for example, for GPT-OSS-20B the oracle achieves 5% higher accuracy while invoking CoT on less than half of the queries.

**Efficiency Gains Vary Across Tasks.**    *Commonsense, soft reasoning, and knowledge tasks benefit the most from confidence-based gating.* In Figure 3, we show representative examples including CSQA and Strategy QA. On datasets such as MMLU, StrategyQA, and MUSR, both GPT-OSS and Qwen3-32B can achieve the same accuracy as always using CoT while reducing token usage by 30–50%. In some cases, such as StrategyQA and MUSR with GPT-OSS and Qwen3-32B, performance even improves slightly at certain budgets while using fewer tokens. Figure 3 also shows high potential for these tasks, with the oracle using about 75% less CoT for CSQA and StrategyQA. In contrast, mathematical and scientific tasks show limited benefit. For GSM8K, direct answering without CoT has very low accuracy, making it difficult to save tokens without hurting performance. This is also clear from the oracle, which shows less room for improvement (Figure 3). Similarly, on GPQA, some confidence methods (e.g., *perplexity* for GPT-OSS-20B) perform better than random and yield modest savings, but the efficiency gains are much less pronounced. The oracle highlights that there is headroom for efficiency on GPQA, but current models are not effective at separating

correct from incorrect answers for these challenging questions. Full results across all datasets can be found in Appendix E.

**No Confidence Method Dominates.** As seen in Figure 2, the effectiveness of confidence signals varies strongly across models. For GPT-OSS-20B, *margin* and *perplexity* can outperform random gating across all budgets. In contrast, both $P(True)$ and *verbalised* confidence often perform worse than random. For Qwen3-32B, $P(True)$ emerges as the most effective method, outperforming other signals across a wide range of budgets. *Margin* and *perplexity* achieve above-random performance only at low budgets, but quickly saturate: the distributions collapse to narrow ranges, limiting their separating power. Finally, for Qwen3-8B, no method consistently outperforms random gating across all budgets.

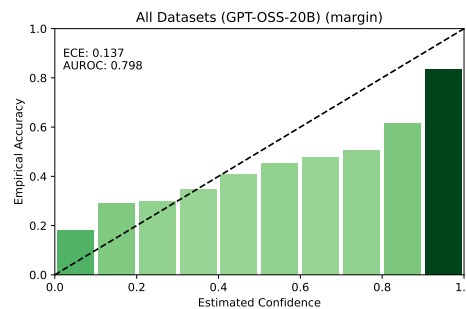

Figure 4: Reliability diagram for GPT-OSS-20B with *margin* confidence.

**Scale and Calibration Effects.** To better understand why confidence gating is more effective in larger models, we look at the calibration of each confidence signal. Figure 4 shows reliability diagrams for GPT-OSS-20B, where margin sampling achieves the highest AUROC. Broadly, both GPT-OSS-20B and Qwen3-32B achieve higher AUROC across methods compared to Qwen3-8B (Appendix F). This suggests that, at a larger scale, LLMs are better calibrated and can more reliably separate correct and incorrect predictions. This finding is consistent with prior findings that calibration improves with model size (Kadavath et al., 2022). It also demonstrates why we observe positive confidence gating results from the larger models. Notably, Qwen3-8B generally produces longer CoT with an average length of 1,269 tokens compared to 625 for GPT-OSS-20B (high) and 884 for Qwen3-32B. Although Qwen3-8B stands to benefit the most from effective gating, its weak calibration prevents it from achieving these gains.

## 5.1 REALISTIC CoT DEPLOYMENT

In realistic settings, a model must decide when to invoke CoT given a budget, without access to the full confidence distribution. As outlined in §3.3, we address this with dynamic percentile thresholding (Ramirez et al., 2024), which updates thresholds online from past scores. We first examine budget–accuracy trade-offs under this setting, and then turn to Pareto-optimal thresholds, which approximate realistic deployment by selecting accuracy-preserving operating points from a calibration set.

**Online CoT Budget–Accuracy Trade-offs.** We implement the dynamic percentile thresholding procedure from §3.3 to enforce CoT budgets in the online setting. Figure 5 shows that the online curves broadly mirror the offline ones, confirming that CoT budgets can remain effective under realistic deployment conditions. The main difference is increased variance. GPT-OSS-20B remains stable across budgets with behaviour very similar to the offline setting. However, for the Qwen3 models,

Table 1: Results for all datasets with Pareto-optimal thresholds ($\epsilon = 1\%$). Accuracy remains within 1% of All CoT; differences are in CoT usage and tokens saved per query.

| | Method | Acc. ↑ | ΔAcc ↑ | CoT (%) ↓ | Avg. Tok. saved ↑ |
|---|---|---|---|---|---|
| **GPT-OSS-20B** | All CoT | 79.9 | 0.0 | 100.0 | 0.0 |
| | All Direct | 54.1 | -25.9 | 0.0 | 483.3 |
| | $P(True)$ | 79.2 ± 0.5 | -0.7 | 95.5 ± 2.3 | 15.3 ± 8.9 |
| | Verbalised | 79.7 ± 0.1 | -0.2 | 99.2 ± 0.0 | 1.1 ± 3.1 |
| | Margin | 79.1 ± 0.4 | -0.8 | **68.1 ± 3.8** | 65.0 ± 12.1 |
| | Perplexity | 78.9 ± 0.5 | -1.0 | 70.6 ± 7.9 | 65.7 ± 22.0 |
| | Oracle | 85.0 | +5.1 | 45.9 | 187.2 |
| **Qwen3-32B** | All CoT | 83.8 | 0.0 | 100.0 | 0.0 |
| | All Direct | 67.8 | -16.0 | 0.0 | 878.6 |
| | $P(True)$ | 82.8 ± 0.5 | -1.0 | **73.8 ± 5.6** | **170.9 ± 45.1** |
| | Verbalised | 83.7 ± 0.1 | -0.1 | 98.9 ± 0.0 | 3.8 ± 4.1 |
| | Margin | 83.8 ± 0.1 | 0.0 | 100.0 ± 0.0 | 0.0 ± 4.1 |
| | Perplexity | 83.8 ± 0.1 | 0.0 | 100.0 ± 0.0 | 0.0 ± 4.1 |
| | Oracle | 87.9 | +4.1 | 32.2 | 446.7 |
| **Qwen3-8B** | All CoT | 79.1 | 0.0 | 100.0 | 0.0 |
| | All Direct | 57.8 | -21.3 | 0.0 | 1265.1 |
| | $P(True)$ | 78.4 ± 0.5 | -0.7 | **90.8 ± 4.9** | **86.6 ± 54.9** |
| | Verbalised | 79.0 ± 0.3 | -0.1 | 100.0 ± 0.5 | 0.2 ± 6.2 |
| | Margin | 79.1 ± 0.2 | 0.0 | 100.0 ± 0.0 | 0.0 ± 5.6 |
| | Perplexity | 79.1 ± 0.2 | 0.0 | 100.0 ± 0.0 | 0.0 ± 5.6 |
| | Oracle | 83.8 | +4.0 | 42.2 | 563.8 |

*margin* and *perplexity* show noticeably higher variability at mid-to-high budgets, reflecting instability from a loss of separability in their scores. In contrast, $P(True)$ on Qwen3-32B remains stable

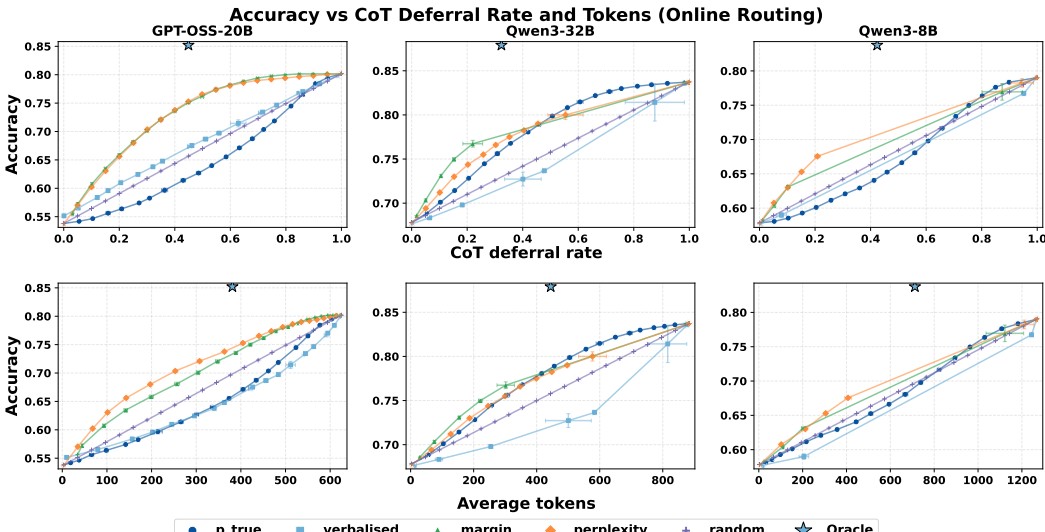

Figure 5: Online Accuracy vs. CoT deferral rate (top) and average tokens (bottom) across all datasets in the **online** setting. Stars show oracle performance.

and very close to its offline performance. Beyond online budget trade-offs, we also consider whether we can find a Pareto-optimal threshold using a calibration set.

**Pareto-optimal thresholds.** We implement the procedure from §3.3 using a 10% calibration split, $\epsilon = 1\%$, and 100 repeats sampling a different calibration split each time (Xu & Liang, 2001). We report the mean and standard deviation of accuracy and cost across these runs. Table 1 reports these results. We see that for Qwen3-8B, $P(True)$ maintains accuracy within 1% of the full CoT baseline while reducing CoT usage by around 10% and saving 89 tokens per query on average. This shows that although no method on Qwen3-8B consistently outperforms the random baseline across the full budget sweep, confidence signals can still identify thresholds that deliver useful savings without hurting accuracy. The larger models also yield Pareto-optimal thresholds that preserve accuracy while lowering token cost, with GPT-OSS-20B achieving reductions of 30-35% in CoT usage and Qwen3-32B showing meaningful savings under $P(True)$. These results confirm that even when the overall trade-off curves appear modest, calibration can highlight settings where performance is maintained and efficiency improves.

## 6 ANALYSIS

To better understand how confidence-gated CoT operates in practice, we examine qualitative examples of both successful and unsuccessful gating cases for a maximum accuracy Pareto-optimal threshold obtained following the method described in the previous section. Then, we separate outcome types from cases where CoT is genuinely needed to those where it adds little or no value (i.e., direct answers that are already correct). Table 2 shows the distribution of these outcomes, defined by the policy's decision to use CoT and the correctness of the final answer.

**Outcome breakdown.** Table 2 reports average outcomes at the Pareto-optimal thresholds selected in the previous results, using the best-performing confidence method for each model. The largest share of cases for Qwen3-8B (50.8%) and Qwen3-32B (44.4%) falls into *Excess CoT*, where the direct answer was already correct but the policy still used reasoning. GPT-OSS-20B is lower at 26.5%. The *Direct* category, where the policy chose to answer directly and that answer was correct, improves with scale, from 7.0% (Qwen3-8B) to 23.5% (Qwen3-32B) and 27.2% (GPT-OSS-20B). This trend is expected since knowing when to use direct requires the model to have well calibrated confidence scores, and Qwen3-8B shows the poorest calibration. *CoT Fixed* accounts for 24.7% of queries on Qwen3-8B, 18.8% on Qwen3-32B, and 29.9% on GPT-OSS-20B, capturing the cases where the direct answer would have been wrong but enabling CoT corrected it. *Both Fail*, cases where gating couldn't help, remain around 12–16% across models. We present representative examples of some of these categories below.

**Example 1: *CoT Fixed*.** Across models, 18–30% of queries fall into this category. Good examples of this come from GSM8K, where direct answering often fails but CoT achieves high accuracy (Sprague et al., 2025). In Example 1, the direct answer was wrong, but the CoT reasoning solved the problem correctly.

> ### Example 1
>
> **Question:** Martha is planning her Christmas party. She invited 2 families with 6 people and 3 families with 4 people. 8 people couldn't come due to illness, and 1/4 that number had previous commitments. How many people show up for Martha's party?
> **Direct answer:** 25 (incorrect).
> **CoT reasoning (excerpt):** "Okay, let me try to figure out how many people are coming to Martha's Christmas party... 2 families times 6 people = 12. 3 families times 4 people = 12. Adding together gives 24 invited. Now, 8 people couldn't come due to illness. Then, 1/4 of that number (2) had previous commitments... So total not coming = 8 + 2 = 10. Therefore, 24 - 10 = 14. ...Yes, that seems correct. **The answer is 14.**"
> **Ground Truth:** 14.

**Example 2: *Direct*.** This represents a positive case where accuracy is preserved and tokens are saved by directly answering. This example saved 284 tokens by choosing to skip CoT.

> ### Example 2
>
> **Question:** Would a Nike shoebox be too small to fit a swan in?
> **Direct answer:** *Yes* ($P(\text{True}) = 0.99$).
> **Ground Truth:** *Yes*.

**Example 3: *Excess CoT*.** In this example, the direct answer was already correct, but the policy still used CoT, leading to redundant tokens.

> ### Example 3
>
> **Question:** Where would you put a glass after drinking from it?
> **Answer choices:** (A) ocean, (B) water cooler, (C) cabinet, (D) dishwasher, (E) dining room.
> **Direct answer:** *(D)* ($P(\text{True}) = 0.59$).
> **CoT reasoning (excerpt):** "Option A doesn't make sense... Option D, dishwasher, is correct. Therefore, the answer is D."

These examples highlight both the promise and the limitations of confidence-gated CoT. On the positive side, gating can recover accuracy when CoT is required (as in GSM8K) and preserve accuracy while saving tokens when direct answers are sufficient. At the same time, unnecessary CoT remains common, with unnecessary reasoning the single largest category in our breakdown (Table 2). This underlines that while training-free confidence signals can guide useful savings, they are inconsistent in practice, and stronger, more consistent gating indicators will be needed for reliable CoT deployment.

Table 2: Distribution of outcome categories across three models. Values are averages over calibration runs with standard deviations shown.

| Category | Qwen8B | Qwen32B | OSS20B |
|---|---|---|---|
| CoT Fixed | 24.7%$_{\pm0.8}$ | 18.8%$_{\pm0.6}$ | 29.9%$_{\pm0.6}$ |
| Direct | 7.0%$_{\pm3.6}$ | 23.5%$_{\pm4.6}$ | 27.2%$_{\pm2.5}$ |
| Excess CoT | 50.8%$_{\pm3.6}$ | 44.4%$_{\pm4.6}$ | 26.5%$_{\pm2.5}$ |
| Missed Fix | 1.2%$_{\pm0.8}$ | 1.3%$_{\pm0.6}$ | 1.3%$_{\pm0.6}$ |
| Both fail | 16.2%$_{\pm0.1}$ | 12.1%$_{\pm0.1}$ | 15.0%$_{\pm0.1}$ |

## 7 CONCLUSION

To our knowledge, we conducted the first systematic study of confidence-guided CoT gating in LLMs. Our results show that training-free confidence signals preserve accuracy and cut redundant reasoning by 25–30%, thereby lowering overall token cost. These findings imply that LLMs already possess useful self-assessment signals that can make reasoning more efficient, especially at scale, but current confidence estimation methods are too brittle for robust deployment. The challenge ahead is to develop models that are not only capable of reasoning but also calibrated in terms of when to reason. Progress would lower inference cost and latency while improving reliability, making adaptive CoT a practical tool for large-scale, real-world systems.

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

## A  REPRODUCIBILITY STATEMENT

Our code to reproduce all experiments is available on an anonymous GitHub repository: https://anonymous.4open.science/r/cgr-DDCE. This repository will remain accessible until the ICLR 2026 decision notification date: Jan 22, 2026 (AOE). All inference hyperparameters are specified in Appendix B. Experiments were run on a combination of Nvidia A100 (80GB) and H100 (80GB) GPUs. We report results as the mean and standard deviation across repeated experiments. We will also provide all generated outputs, including CoT traces and confidence scores. All datasets are publicly available.

## B  MODEL INFERENCE SETTINGS

We use Hugging Face Transformers for all inference. For Qwen models (8B and 32B), we follow the recommended decoding settings from the model cards, using temperature 0.6 and top-p 0.95 to avoid degenerate repetition. For GPT-OSS-20B, we use the default sampling configuration with temperature 1.0 and top-p 1.0. In all setting we set a maximum limit of 7000 thinking tokens and insert text that prompts the model to answer after this limit has been reached.

## C  LLM USAGE

The writing of this paper received proofreading and language polishing suggestions using LLMs. In addition, parts of our experimental code were drafted or refactored with the assistance of GitHub Copilot; all final text and code was manually reviewed and verified by the authors.

## D  PROMPTS AND DATASET STATISTICS

> **Verbalised Prompt**
>
> Please directly provide your best guess of the answer to the question and give the probability that you think it is correct (0.0 to 1.0). Take your uncertainty in the prompt, the task difficulty, your knowledge availability, and other sources of uncertainty into account.
> Give only the guess and probability, with no other words or explanation.
>
> Format your final response as:
> Answer: <your_best_guess>.
> Probability: <score between 0.0 and 1.0>

> **$P(\text{True})$ Prompt**
>
> User:
> Is this answer:
> (A) True
> (B) False
>
> Assistant:
> The answer is:

Table 3: Dataset statistics.

| Dataset | # Samples |
|---|---|
| CommonsenseQA (CSQA) | 1221 |
| StrategyQA | 2290 |
| MMLU-redux | 3000 |
| GSM8K | 1319 |
| GPQA | 448 |
| LSAT-AGI | 1009 |
| MUSR | 756 |

## E  PER DATASET TRADE-OFF PLOTS

Figure 6: CSQA: Accuracy vs. CoT use (top) and average tokens (bottom) across models

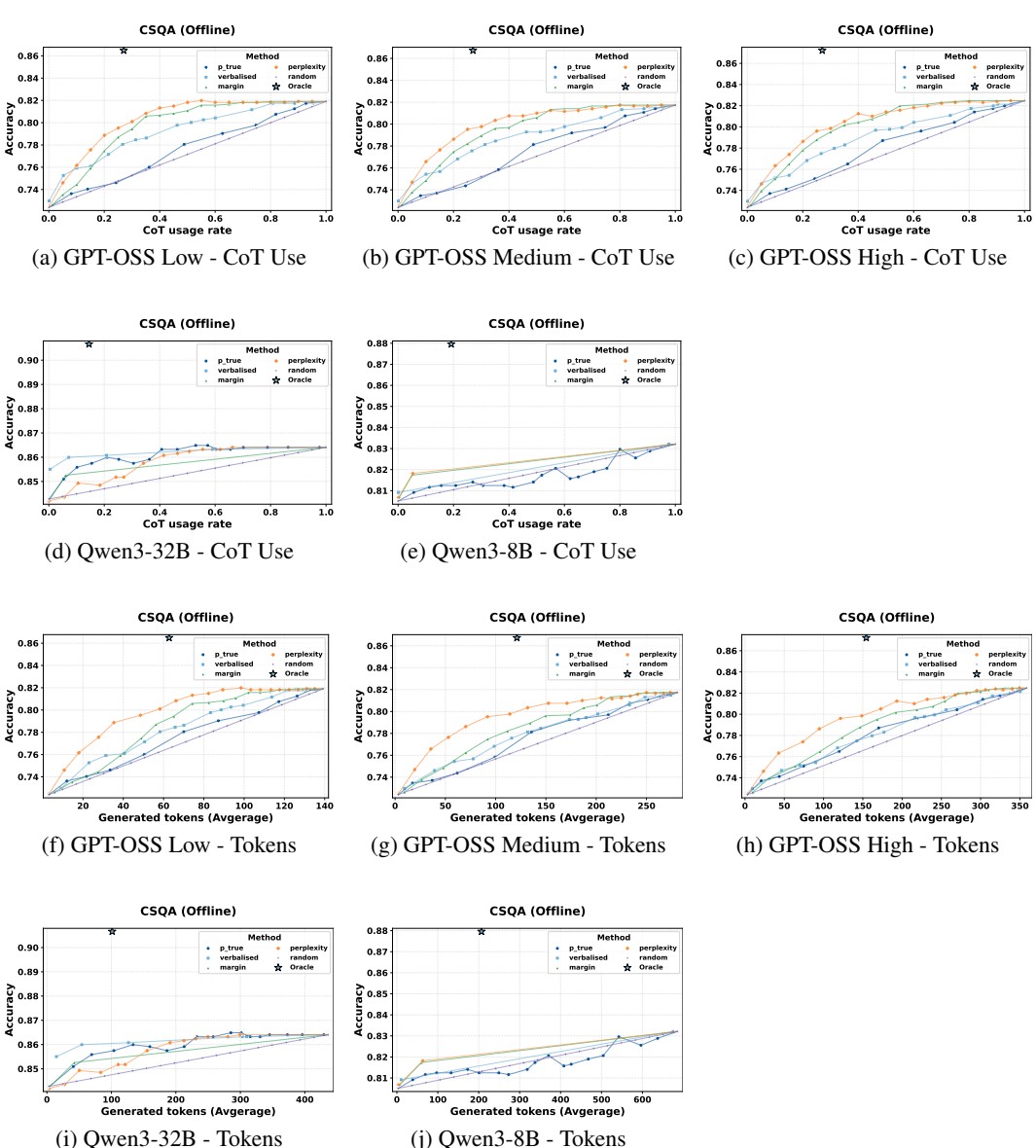

(a) GPT-OSS Low - CoT Use        (b) GPT-OSS Medium - CoT Use        (c) GPT-OSS High - CoT Use

(d) Qwen3-32B - CoT Use        (e) Qwen3-8B - CoT Use

(f) GPT-OSS Low - Tokens        (g) GPT-OSS Medium - Tokens        (h) GPT-OSS High - Tokens

(i) Qwen3-32B - Tokens        (j) Qwen3-8B - Tokens

Figure 7: GPQA: Accuracy vs. CoT use (top) and average tokens (bottom) across models

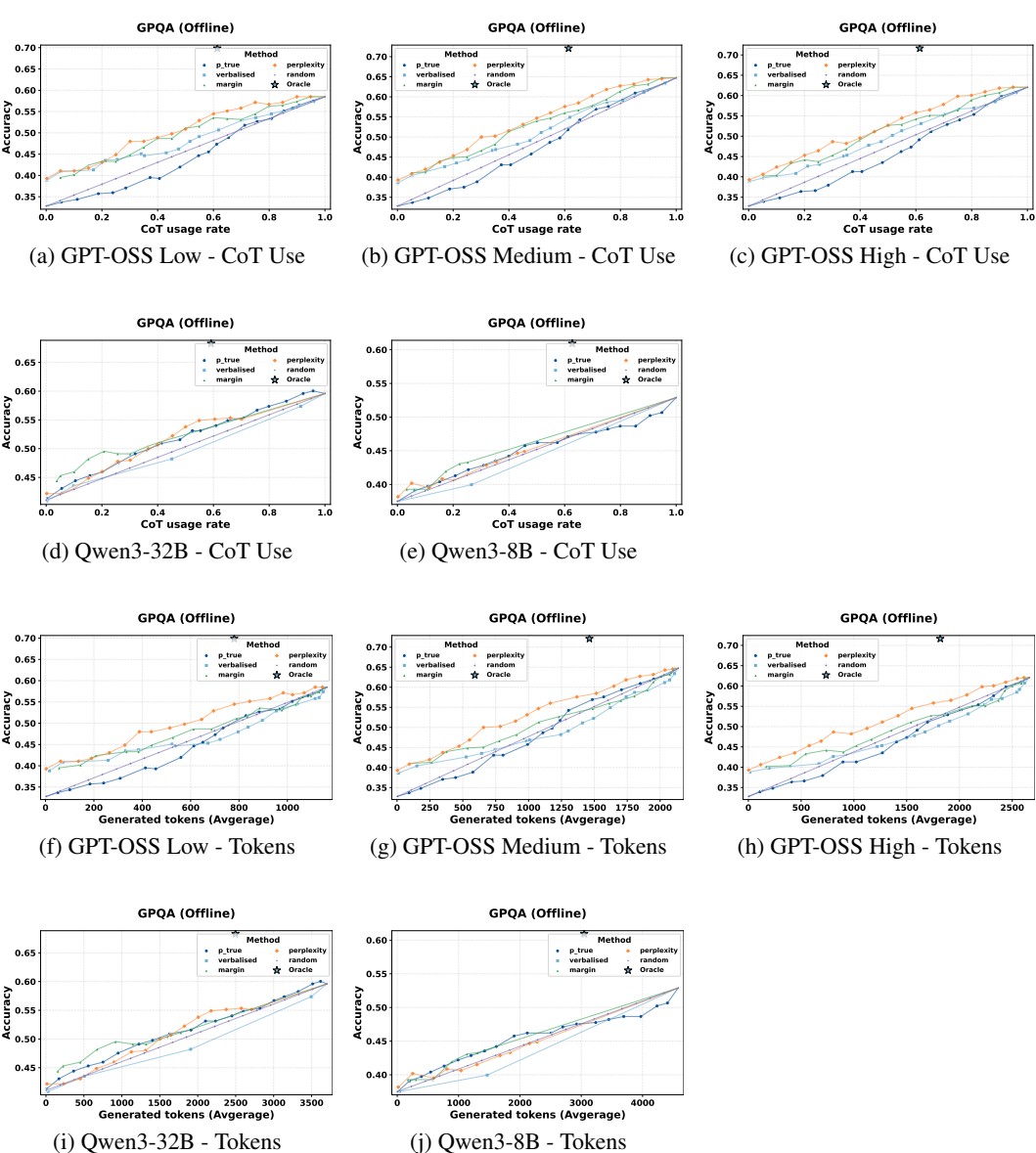

(a) GPT-OSS Low - CoT Use

(b) GPT-OSS Medium - CoT Use

(c) GPT-OSS High - CoT Use

(d) Qwen3-32B - CoT Use

(e) Qwen3-8B - CoT Use

(f) GPT-OSS Low - Tokens

(g) GPT-OSS Medium - Tokens

(h) GPT-OSS High - Tokens

(i) Qwen3-32B - Tokens

(j) Qwen3-8B - Tokens

Figure 8: GSM8K: Accuracy vs. CoT use (top) and average tokens (bottom) across models

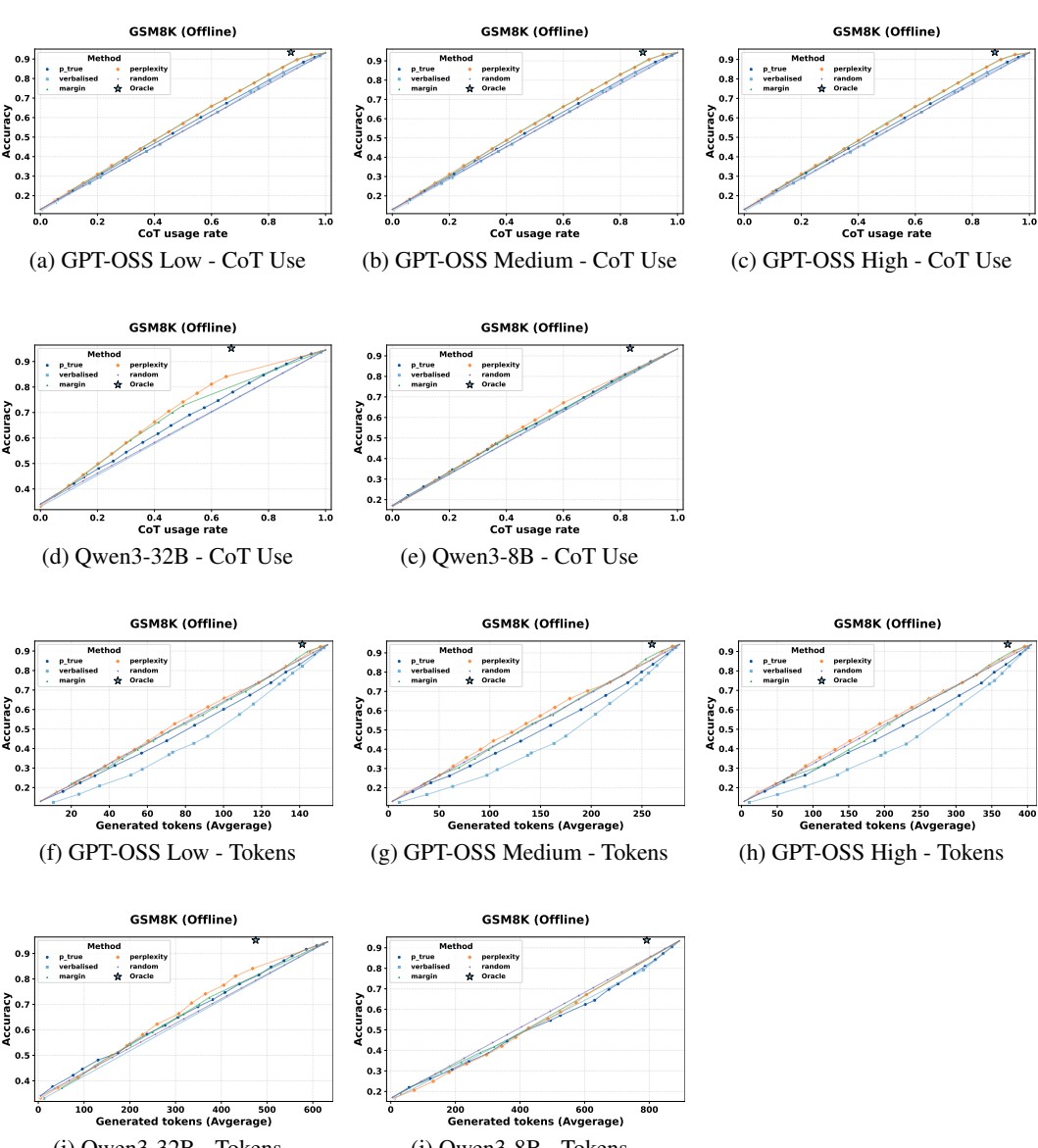

Figure 9: LSAT-All: Accuracy vs. CoT use (top) and average tokens (bottom) across models

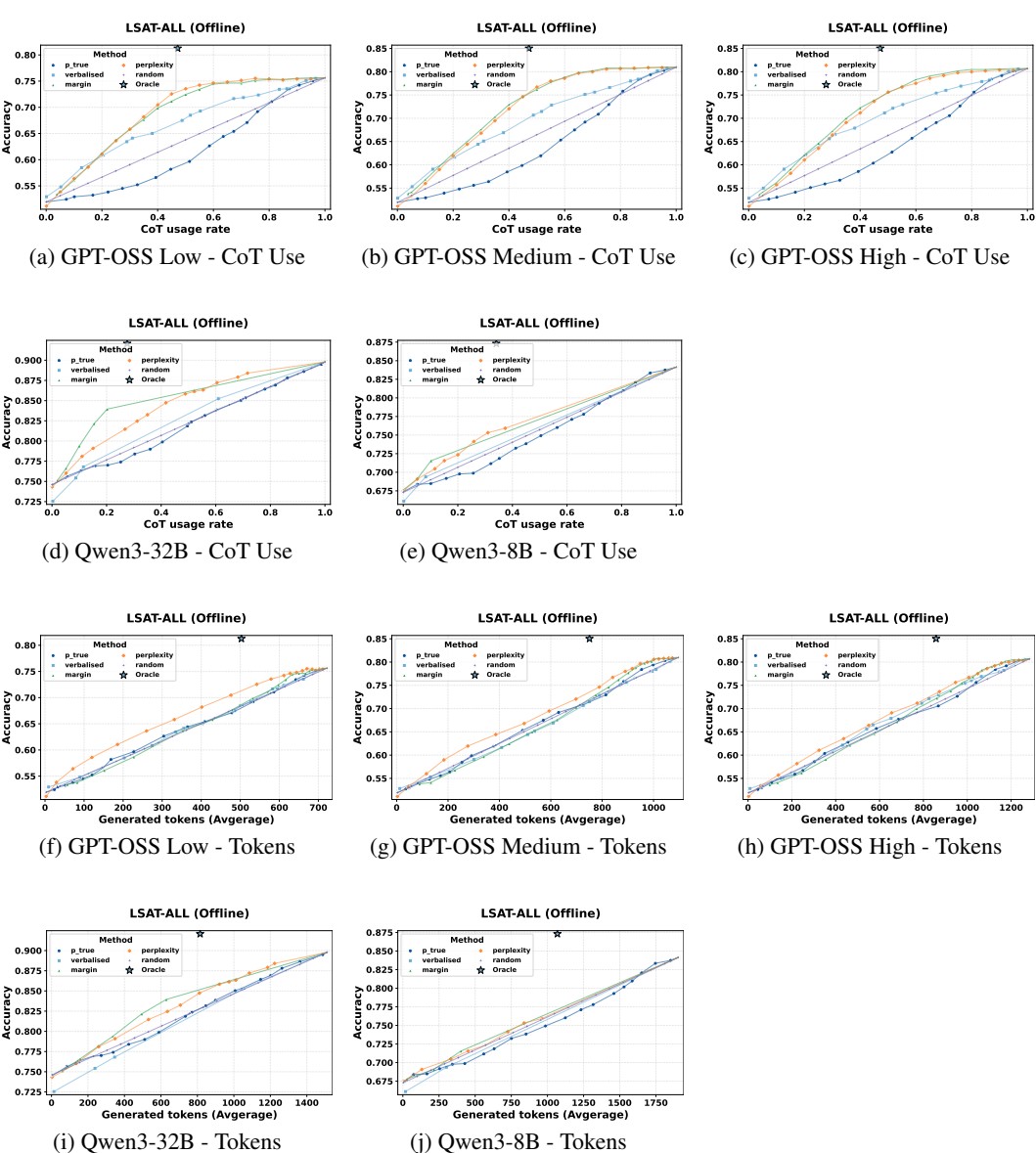

(a) GPT-OSS Low - CoT Use

(b) GPT-OSS Medium - CoT Use

(c) GPT-OSS High - CoT Use

(d) Qwen3-32B - CoT Use

(e) Qwen3-8B - CoT Use

(f) GPT-OSS Low - Tokens

(g) GPT-OSS Medium - Tokens

(h) GPT-OSS High - Tokens

(i) Qwen3-32B - Tokens

(j) Qwen3-8B - Tokens

Figure 10: MuSR-All: Accuracy vs. CoT use (top) and average tokens (bottom) across models

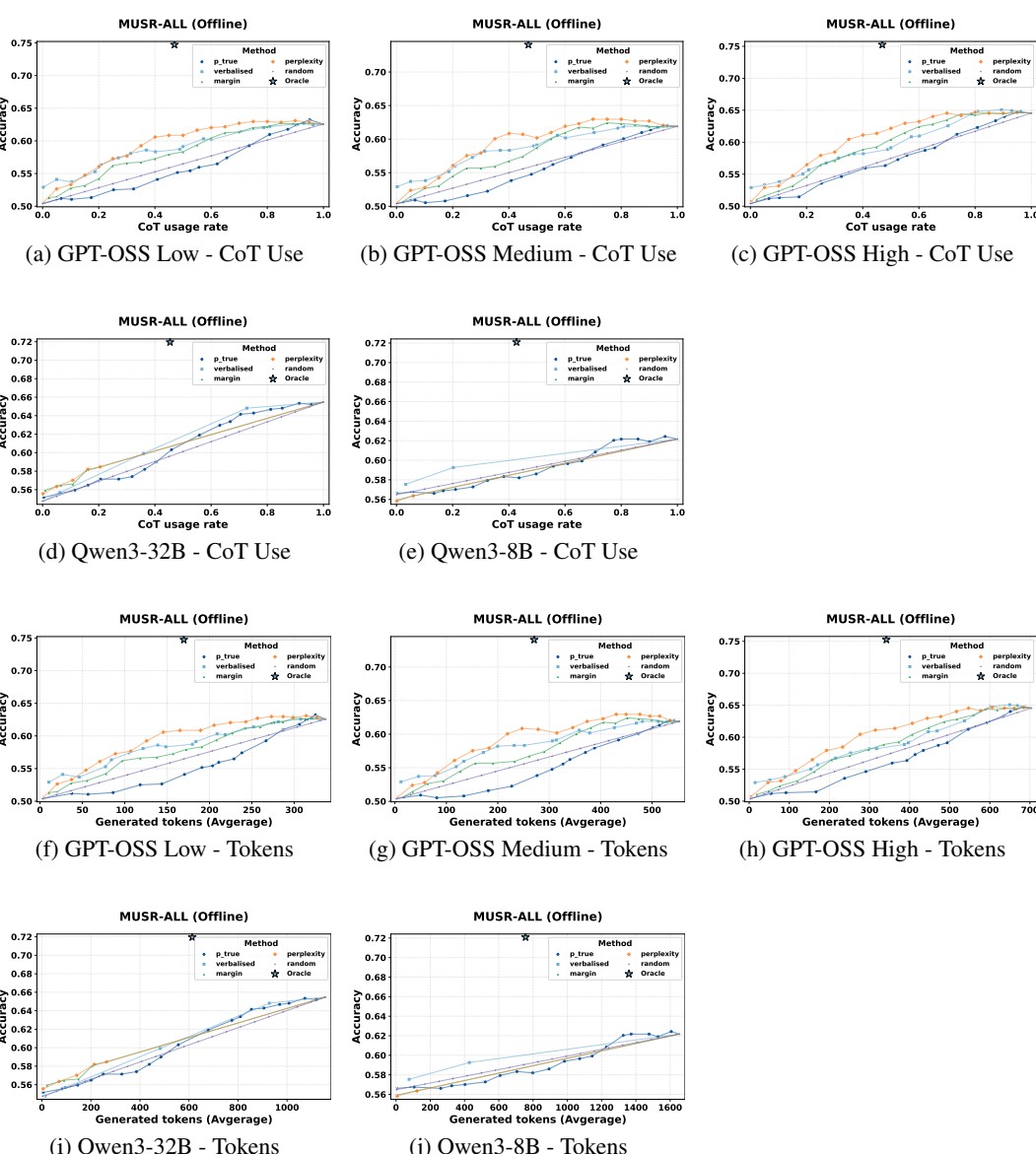

Figure 11: MMLU-Redux: Accuracy vs. CoT use (top) and average tokens (bottom) across models

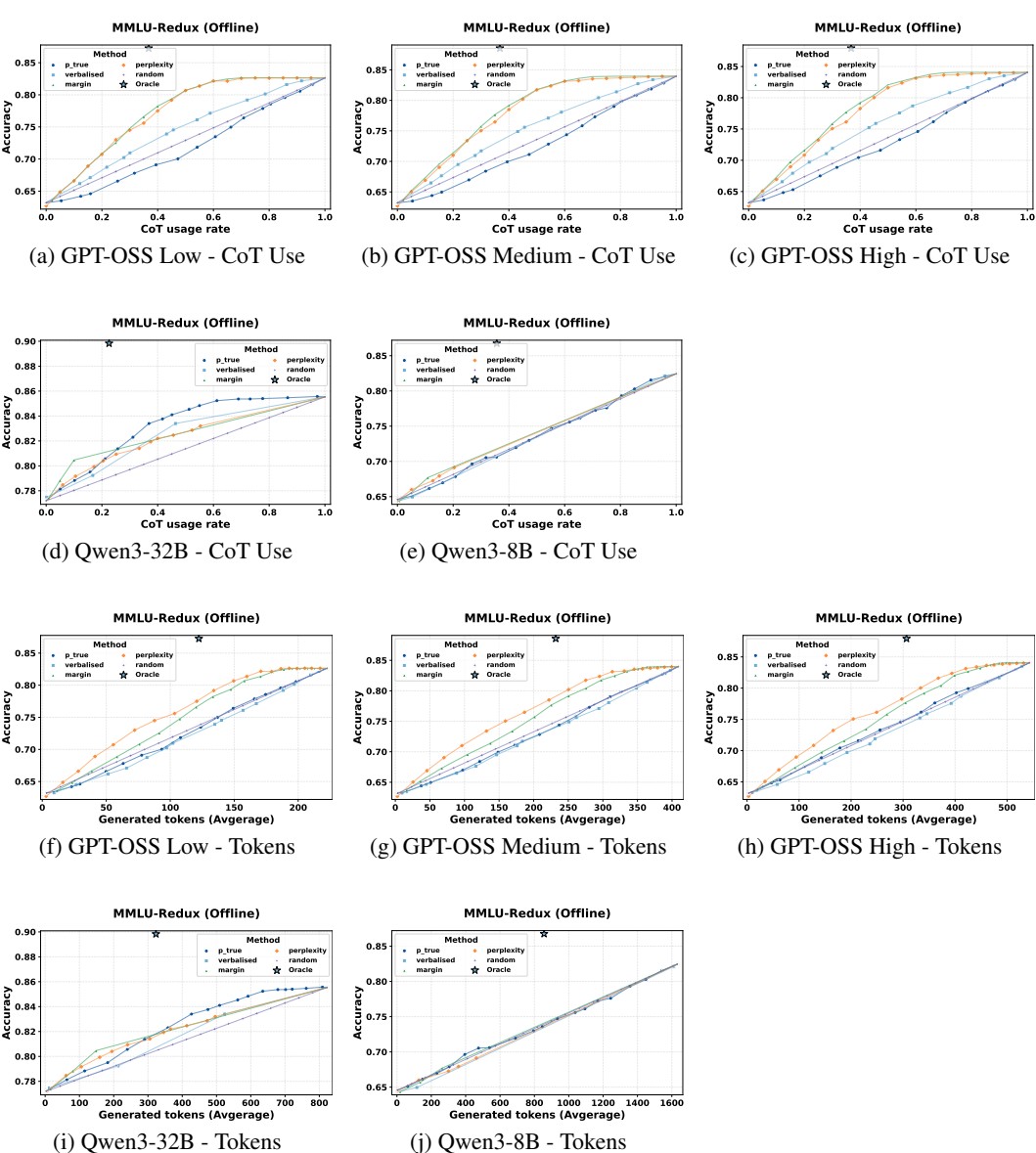

(a) GPT-OSS Low - CoT Use     (b) GPT-OSS Medium - CoT Use     (c) GPT-OSS High - CoT Use

(d) Qwen3-32B - CoT Use     (e) Qwen3-8B - CoT Use

(f) GPT-OSS Low - Tokens     (g) GPT-OSS Medium - Tokens     (h) GPT-OSS High - Tokens

(i) Qwen3-32B - Tokens     (j) Qwen3-8B - Tokens

Figure 12: StrategyQA: Accuracy vs. CoT use (top) and average tokens (bottom) across models

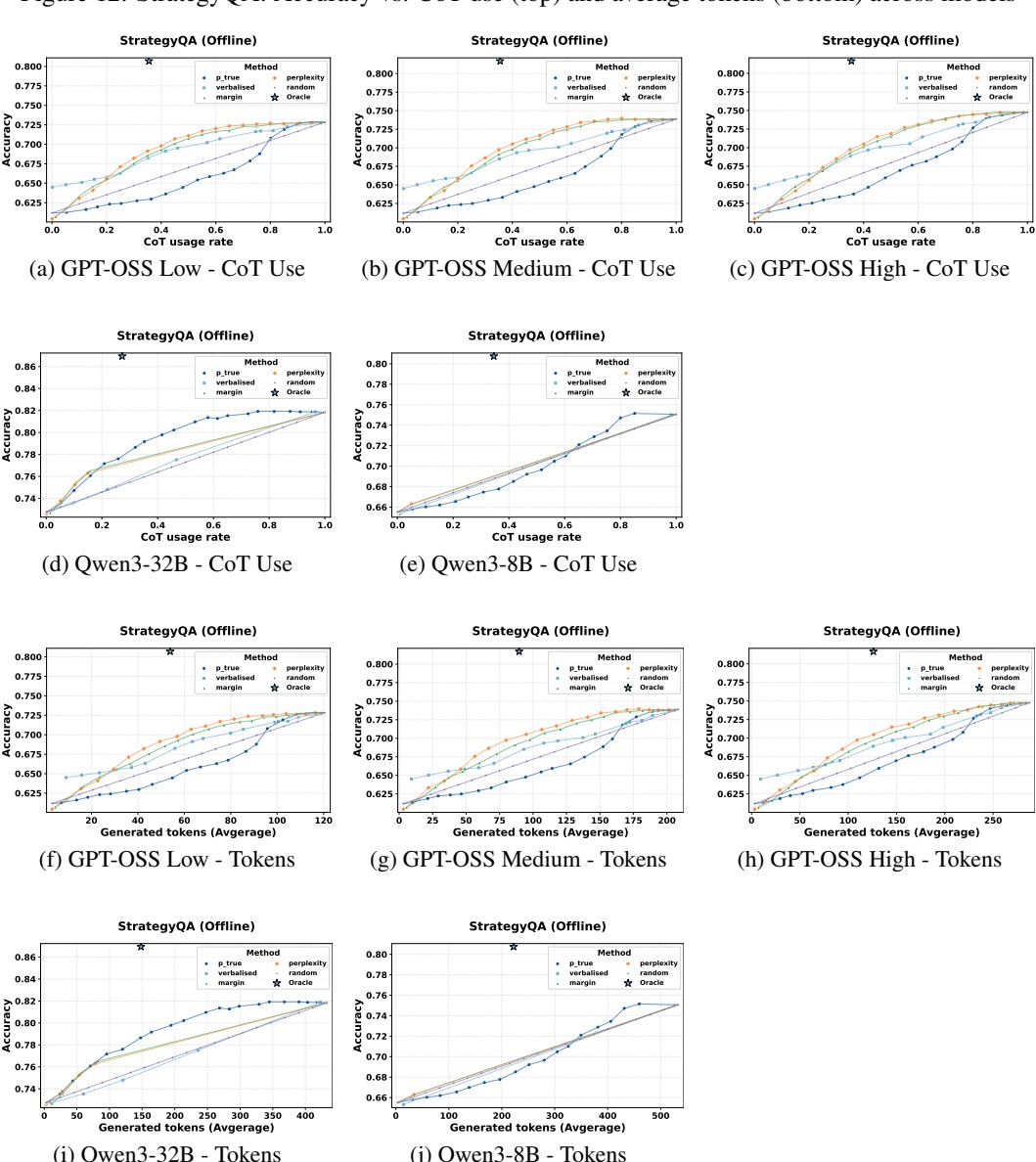

## F    RELIABILITY DIAGRAMS

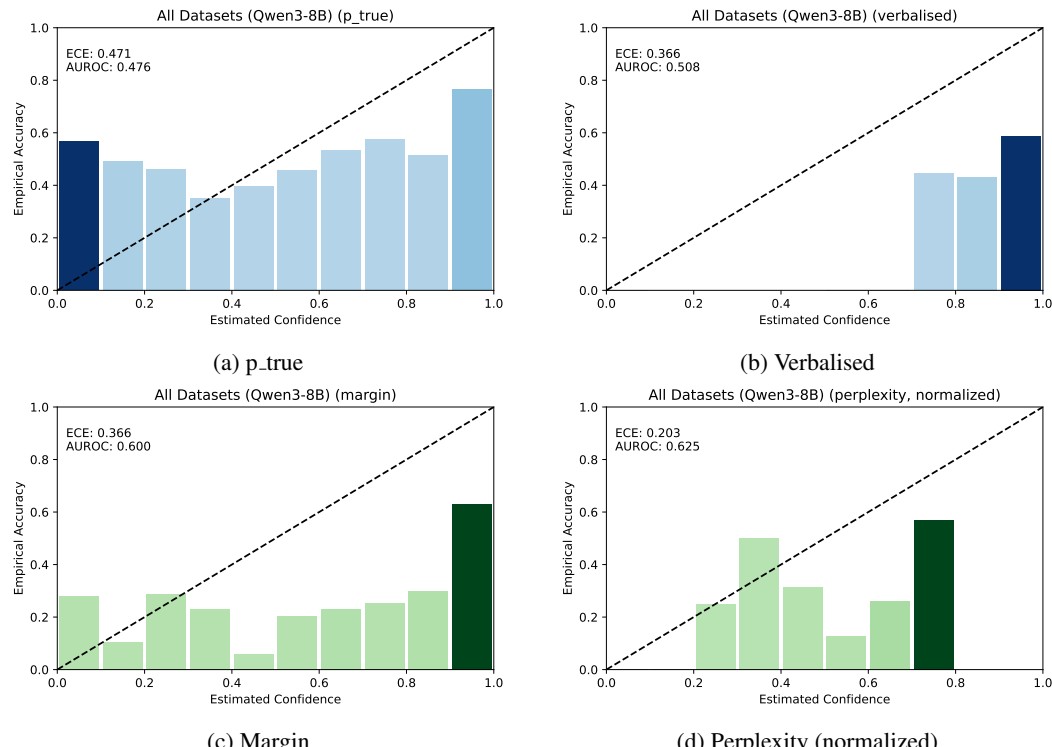

(a) p_true

(b) Verbalised

(c) Margin

(d) Perplexity (normalized)

Figure 13: Reliability diagrams for **Qwen3-8B**. Bars darken with bin count; dashed line is perfect calibration.

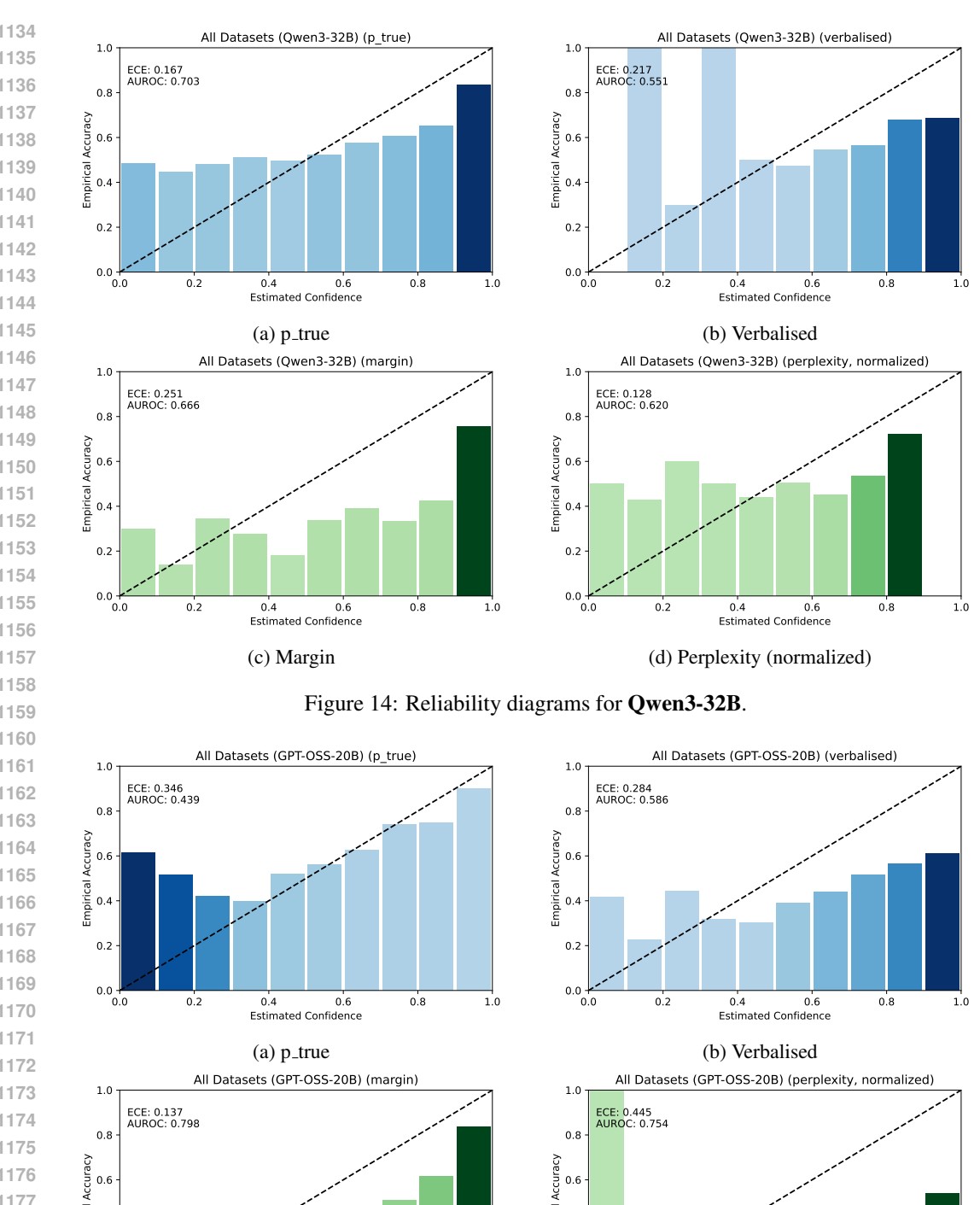

Figure 14: Reliability diagrams for **Qwen3-32B**.

Figure 15: Reliability diagrams for **GPT-OSS-20B**.

