# OpenReview forum: "Can Confidence Estimates Decide When Chain-of-Thought is Necessary for LLMs?"
_ICLR.cc/2026/Conference — ICLR 2026 Conference Withdrawn Submission_

### Official Review · Reviewer_3F3L · 2025-10-25

**Soundness:** 1
**Presentation:** 3
**Contribution:** 1
**Rating:** 2
**Confidence:** 4

**Summary:**

This paper introduces and analyzes a confidence-gating method for dynamically deciding when CoT reasoning is necessary for LLMs. Specifically, the model first generates a direct answer and then estimates its own confidence using training-free signals, such as perplexity, P(True), margin sampling, and verbalized confidence. Based on this confidence score, a gating threshold determines whether to accept the direct answer or to invoke CoT. Experiments are conducted across models and tasks to discuss the effectiveness and limitations of the idea of using confidence to determine the use of CoT.

**Strengths:**

1. In general, the writing of the paper is clear and easy to follow.
2. The focus of the paper, on how to determine the necessity of using CoT, is meaningful and interesting.
3. Multiple tasks are considered in the experiment section, which provides extensive results on how confidence-gated CoT behaves across different reasoning types and model scales.

**Weaknesses:**

1. A major concern is that the positioning of the paper is somewhat unclear.
While it introduces a method for confidence-based control of CoT reasoning, the proposed approach fails to consistently demonstrate strong performance across models and tasks. As a result, the paper shifts its focus toward analyzing whether confidence estimates can be used to decide when CoT is needed. However, since the analysis is conducted only on this specific implementation, the conclusions are method-specific and lack generality or broader applicability.
2. The novelty of the work might be limited. The idea of applying confidence to improve the efficiency of reasoning during inference has been applied and analyzed in multiple works [1][2][3]. While the implementation here is not identical to previous approaches, the conceptual contribution appears incremental, and the methodological design does not seem to introduce substantial innovation beyond existing frameworks.
[1] Prolonged Reasoning Is Not All You Need: Certainty-Based Adaptive Routing for Efficient LLM/MLLM Reasoning
[2] Stepwise Perplexity-Guided Refinement for Efficient Chain-of-Thought Reasoning in Large Language Models
[3] Uncertainty-Guided Chain-of-Thought for Code Generation with LLMs
3. In some experiment sections, the authors mainly describe the results of the experiments without providing a comprehensive analysis to discuss the reasons behind the phenomenon and the corresponding insights. For example, why does the performance of the confidence-gating method perform better in certain tasks but worse in others? Why do P(true) and verbalized confidence sometimes (even with capable models) perform worse than randomly deciding the usage of CoT?

**Questions:**

1. In the paper, the authors mentioned that a calibration set is used for selecting the optimal threshold. Given that a calibration set can be used, why is it necessary to develop an online evaluation? Is there a clear comparison between the calibration-based Pareto threshold and the dynamic percentile thresholding?
2. Why do some lines in the trade-off plots (e.g., perplexity or margin confidence) contain very few points, especially at higher CoT-usage rates?

---

> ### Author Response · Authors · 2025-11-18
>
> Thank you for your review and engagement. We provide our response to the points raised below.
>
> **W1: Unclear positioning of the paper**
> We appreciate your comment on the framing of the paper. However, we would be grateful *if you could clarify what you mean by “method-specific” and the analysis being conducted only on “this specific implementation”*. **Our analysis does not rely on a single implementation**. We provide “extensive“ (Reviewer 3F3L), “systematic and broad” (Reviewer 7N6a) results by evaluating four different confidence signals, across model families, model scales, and reasoning types, and under budget-constrained, accuracy-preserving, and both offline and online decision settings. This broad experimental setup and framework allow us to move beyond any individual mechanism and identify general trends about when confidence-gated CoT is effective and when it is not across a wide range of different settings. This is an underserved area of reasoning efficiency, and our work provides a strong empirical foundation and evaluation framework needed for the community to build on.
>
> **W2: The novelty of the work might be limited.**
> Prior work has used confidence to improve reasoning efficiency. However, our method provides a unique evaluation framework and extensive empirical results addressing a critical gap in the literature. The dominant paradigm focuses on shortening CoT after it has already begun, as in [2]. This leaves a major open question unaddressed: can confidence be used to decide if CoT should be used at all? Although [1] also considers confidence before reasoning, it does so on a narrow set of tasks and only on one confidence method (perplexity). Similarly, [3] makes line-by-line confidence decisions within code generation, which is not a realistic setting for most reasoning tasks. Therefore, it is unclear if these results generalise to other tasks. As discussed above, our work fills this gap by providing “extensive” empirical evidence on how multiple confidence signals behave across tasks, models, and decision settings, together with the evaluation framework needed to study this problem systematically.
>
> **W3: Analysis to discuss the reasons behind the phenomenon**
> Thank you for the feedback here. We would like to highlight that a discussion regarding why confidence-gating “perform better in certain tasks but worse in others” is provided starting line 313. We would be happy to clarify this further if needed. We agree that an analysis of why some confidence methods perform worse than random (even for strong models) would be valuable. **As such, we will include the following in the updated manuscript starting at the end of line 351:**
>
> > Interestingly, even though larger models are generally better, some methods often perform worse than random e.g. p(true) for gpt-oss and verbalised for Qwen3-8B. This is due to poor discriminative power indicated by low AUROC scores in these instances (below 0.5). This is consistent with prior work which generally finds inconsistent results with verbalised confidence across models (Yang et al., Tian et al.).
>
> We would also like to highlight that we provide an analysis of model size impact starting line 342 and a detailed breakdown of how confidence-gated reasoning hurts and helps in Section 6.
>
> ---
>
> **Question 1: Difference between online and Pareto-optimal setting**
> We agree that the role of the online and offline evaluation could be clarified further. The Pareto-optimal threshold and the budget-based evaluation address different questions. The Pareto setting tests whether accuracy can be preserved while reducing cost, allowing us to see how each confidence method behaves under different CoT-use budgets. Within the budget setting, the offline and online modes capture two deployment scenarios. Offline assumes access to the full confidence distribution (batch setting), while online simulates streaming inputs where thresholds must be updated on the fly. As such, it is not appropriate to compare the online-budget evaluation and the Pareto setting, because they evaluate different objectives and are designed to model different practical constraints. **We will clarify this in the paper with the following:**
>
> > *(At the start of Section 3.3)*
> > At a high level, we evaluate confidence-gated CoT under two distinct setups: a budget-based analysis and a separate Pareto-optimal analysis. …
>
> > *(Line 210)*
> > Online vs Offline Budget Evaluation. For the CoT-use budget analysis, we consider both offline and online settings for estimating percentile thresholds. …
>
> We are happy to clarify this further if needed.
>
> **Question 2: Some lines contain few datapoints in figures**
> Please refer to shared response “Lack of or missing datapoints”.
>
> ---
>
> **References**
>
> [1] *Prolonged Reasoning Is Not All You Need.*
> [2] *Stepwise Perplexity-Guided Refinement for Efficient Chain-of-Thought Reasoning.*
> [3] *Uncertainty-Guided Chain-of-Thought for Code Generation.*

---

### Official Review · Reviewer_7N6a · 2025-10-31

**Soundness:** 2
**Presentation:** 3
**Contribution:** 2
**Rating:** 2
**Confidence:** 4

**Summary:**

This paper studies confidence-gated CoT, a simple method that lets LLMs decide when to use reasoning based on training-free confidence signals. Using GPT-OSS and Qwen3 models, the authors show that these signals can cut unnecessary CoT usage and token cost while keeping accuracy nearly unchanged. However, the results are uneven across models and tasks, showing promise but also limitations in the reliability of the current confidence measure.

**Strengths:**

1. The paper tackles a clear and practical problem, deciding when CoT reasoning is necessary through an intuitive confidence-based gating framework.
2. The experiments are systematic and broad, covering multiple models and reasoning benchmarks with solid comparisons to random and oracle baselines.
3. Results demonstrate meaningful token savings with minimal accuracy loss, showing practical value for efficient reasoning.

**Weaknesses:**

1. I am concerned that the contribution of this paper is limited. If the goal is to propose an efficient reasoning method, then the performance is not good enough and there are many efficient CoT baselines to be compared with. If the goal is to simply analyze the confidence-gating, then the scope is way too narrow, as effectively measuring the confidence of LLM generation itself is still a problem to be solved. The insights from this paper are limited.

2. The proposed confidence signals exhibit substantial variance across datasets and model scales, suggesting that their reliability depends largely on the underlying calibration quality of each model rather than on any consistent principle of confidence estimation. This is critical as if wrong confidence is used, the confidence-gating is even worse. This restricts the practical usage of confidence-gating methods.

3. The generalization of the thresholding method is questionable. Selecting the Pareto-optimal threshold relies on the calibration dataset, which may not generalize well to out-of-distribution data. This further indicates that the confidence-gating method studied in this work is far from mature and practical.

**Questions:**

See weakness.

---

> ### Author Response · Authors · 2025-11-18
>
> Thank you very much for your review. We respond to each point below.
>
> **W1: Contribution concern**
> We thank the reviewer for the comment. Our contribution is to establish the first systematic framework for evaluating *training-free, self-assessed confidence signals* for CoT gating. Our goal is not to propose a single “efficient reasoning method” but to provide an empirical and methodological foundation to explore how *confidence* can decide when CoT should be triggered.
>
> Regarding baselines, prior work on efficient CoT focuses primarily on shortening or truncating CoT once it has begun. These are valuable but they address a different decision point. Our work tackles the complementary problem of *whether reasoning should be invoked at all*. As such these methods are better viewed as complementary and can be applied once CoT is triggered by confidence gating, rather than direct baselines for our setting.
>
> As you noted, we observe “meaningful token savings with minimal accuracy loss”. However, other settings show weaker gains, and this contrast is absent from narrower previous studies [1], becoming visible only through our extensive experimental setup.
>
> Beyond these “extensive“ (Reviewer 3F3L), “systematic and broad” (Reviewer 7N6a) empirical results, we introduce a framework for evaluating confidence-gated CoT or any gated CoT method. By including offline/online budget evaluation and a Pareto optimal setup, we provide a clear way to understand how these methods behave under different practical constraints, such as fixed budgets, sequential queries, batch settings, and settings where accuracy preservation is a priority. This is an underexplored avenue for improving reasoning efficiency, and having a clear framework and strong empirical results is essential for the community to build on. Our work provides this foundation.
>
> **W2: Confidence signals exhibit substantial variance**
> The reviewer comments that the “variance across datasets and model scales” may limit direct application. We view these differences across models and scales as informative rather than purely limiting. Our “systematic and broad” experiments show both where confidence-gating already provides value and where and how current methods need improvement. While not all confidence estimation methods work in all scenarios, we observe meaningful token savings with minimal accuracy loss, demonstrating practical value in many settings.
>
> **W3: Generalization of the thresholding method**
> The reviewer raises an important point about OOD generalisation. Our original setup uses a calibration split that is drawn from all datasets. To investigate how well thresholds transfer across tasks, we ran an additional experiment where each dataset is used as a calibration set while the others act as test sets, both individually and jointly. We sampled the same number of examples from each calibration dataset.
>
> More challenging and diverse datasets, such as *MMLU-Redux* and *GPQA*, can provide similar efficiency gains to the calibration split drawn from the same dataset. For example, calibrating P(True) on MMLU-Redux gives a drop of only 0.8 per cent across all other datasets while still reducing CoT usage to 85%. In contrast, datasets like *CSQA* consistently overestimate how much CoT can be removed and produce larger accuracy drops. This is because CSQA questions are significantly easier than many of the other datasets, which leads to thresholds that are too low for the more difficult datasets. Overall, this shows that calibration sets need to reflect the difficulty and diversity of the test tasks. When they do, the Pareto method transfers well and gives stable efficiency gains. When they don’t, performance drops. We will add the full results to the paper's appendix.
>
> **Qwen3-32B Sample results. One dataset is used as calibration, and all others are used as test**
> | Calibration Dataset    | Method   | Acc. ↑ | ΔAcc ↑ | CoT (%) ↓ | Avg. Tok. Saved ↑ |
> |------------------------|----------|--------|--------|-----------|--------------------|
> | CSQA                   | P(True)  | 70.4   | -13.0  | 19.5      | 746.7             |
> | MMLU-Redux             | P(True)  | 82.2   | -0.8   | 84.8      | 116.9             |
>
> We would also like to highlight that the intention of this work was not to provide a “mature” method that works in all scenarios. Our goal is to provide the broad empirical results and an evaluation framework for an underexplored area of CoT efficiency. At the same time, we do not view the method as “far from mature and practical”. As you also note, we do “demonstrate meaningful token savings with minimal accuracy loss, showing practical value for efficient reasoning.”
>
> ---
>
> **References**
> [1] *Prolonged Reasoning Is Not All You Need: Certainty-Based Adaptive Routing for Efficient LLM/MLLM Reasoning*

---

### Official Review · Reviewer_cehT · 2025-11-01

**Soundness:** 2
**Presentation:** 2
**Contribution:** 2
**Rating:** 4
**Confidence:** 4

**Summary:**

This paper investigates whether the model confidence can decide when to invoke chain-of-thought (CoT) reasoning in large language models (LLMs). The authors study a training-free, confidence-gated CoT framework that answers directly when confidence is high and triggers CoT when confidence is low. Through offline and online experiments across multiple models and datasets, they show that confidence-gated CoT achieves the best trade-off between  accuracy and inference cost (in terms of tokens). The analysis also examines calibration (ECE/AUROC) of several confidence signals—verbalized scores, normalized perplexity, margin, and P(True)—and identifies strengths and failure modes.

**Strengths:**

- The paper addresses a timely and relevant problem and is supported by solid empirical evaluations.
- The writing is clear, well-organized, and easy to follow.

**Weaknesses:**

- **Oracle and CoT routing interpretation** In Figures 2 and 3, the oracle triggers CoT only when a direct answer is incorrect (line 286). However, the gap between “always using CoT” and the oracle is very large, suggesting that CoT severely harms performance when applied indiscriminately. This raises a key question: does the oracle use only *one round* of CoT, matching other baselines? If so, the results may indicate that *offline CoT routing itself is ineffective*, which deserves clearer discussion.
- **Claim vs. observed trends** Line 313 states that “Commonsense, soft reasoning, and knowledge tasks benefit the most from confidence-based gating.” However, Appendix E shows that many soft-reasoning and knowledge tasks do *not* exhibit strong gains, and several curves (e.g., Figure 9) resemble math-like trends rather than commonsense improvements. Further analysis is needed to justify the claim and explain task-dependent behaviors.
- **Uncalibrated confidence undermines core comparisons** The main results rely on confidence-based thresholding, yet verbalized confidence is known to be poorly calibrated in LLMs without additional training. If model confidence is uncalibrated, thresholding becomes nearly equivalent to random selection, making the “random gating” baseline an insufficient point of comparison. To support the central claims, the paper should include results with *calibrated* confidence (e.g., confidence tuning, post-hoc calibration, or temperature scaling).
- **Lack of practical guidance for deployment** Although confidence-based gating is evaluated extensively, the results vary significantly across tasks and models, making the method difficult to operationalize in real-world or online settings. The paper would be substantially strengthened by offering *practical guidelines*—e.g., which confidence metric works best for which task types, model families, or prompt formats.
- **External validity beyond hybrid CoT models** The experiments are limited to hybrid “thinking-enabled” models. It remains unclear whether the proposed gating strategy generalizes to standard LLMs where CoT is elicited only through prompting. Ablations using different prompting-based CoT elicitation methods would help establish broader applicability.

**Questions:**

- **Missing data points in plots** In Figures 2 and 3, several points for the *margin* (orange) and *perplexity* (green) curves appear to be missing. Please clarify whether these are absent due to filtering, instability, or another implementation detail.

---

> ### Author Response · Authors · 2025-11-18
>
> Thank you for your review. Below, we provide our response to the points raised.
>
> **W1: Oracle and CoT routing interpretation**
> This is an important and unintuitive result. Yes, the oracle uses the same single-pass CoT generation as the other baselines. The large gap between “Always CoT” and the oracle reflects a known property of current models rather than an issue with our setup. Prior work shows that CoT is not uniformly helpful [1] and can reduce accuracy on queries that the model would otherwise answer correctly with a direct response. Our results are consistent with this: unnecessary CoT sometimes introduces errors, so using it indiscriminately performs worse than selective routing. We will clarify this behaviour in the paper.
>
> **W2: Claim vs. observed trends**
> This observation was intended to describe the trend on larger models, where commonsense and softer reasoning tasks show the clearest gains. As Appendix E shows, Qwen3-8b does not follow this pattern. However, this is largely due to the poor overall performance of the smaller model. We provide a full discussion of this starting on line 342.
>
> **W3: Uncalibrated confidence undermines core comparisons**
> This is an important point to raise. However, our thresholding is based on percentiles, which depend only on the ordering of confidence scores rather than their absolute values. Standard calibration methods such as temperature scaling are monotonic transformations and therefore do not change this ordering. As a result, applying these calibration methods would not affect the percentile thresholds, budget curves, or Pareto fronts. In other words, the behaviour we observe is driven by the ranking quality of each confidence signal. **We will clarify in the paper why we do not use post-hoc calibration methods as follows:**
>
> > As we use percentile thresholds, standard post-hoc calibration methods are not helpful as they only adjust the numerical confidence values but preserve the ranking. Percentile thresholds only depend on the ranking, so these methods would have no effect on the outcome of experiments.
>
> **W4: Lack of practical guidance for deployment**
> This is a great suggestion. As such, we have condensed our results into practical advice and **will add the following to the manuscript:**
>
> > **Practical Guidance for Deployment**
> > Our empirical results allow us to offer practical guidance for applying confidence-gated CoT. AUROC provides a useful early indicator of whether a confidence signal is suitable for routing, since a high AUROC corresponds to a more reliable ranking of correct and incorrect direct answers. Although larger models generally produce stronger signals, it is important to evaluate several confidence measures, as different confidence methods perform differently across different models. Margin is most appropriate for short-answer tasks, while p(true) and verbalised confidence apply more broadly when their ranking quality is sufficient. This guidance can be applied in both the budget and the Pareto optimal setup, depending on the end goal. In addition, our results show that online budgeting performs similarly to offline budgeting, suggesting that confidence-gated CoT can be deployed in streaming or sequential settings without substantial loss of performance. Finally, for tasks where CoT provides a large boost in accuracy (e.g., mathematical reasoning), methods that shorten or refine the generated CoT are likely more appropriate than gating, since the model genuinely needs the reasoning to solve the task.
>
> **W5: External validity beyond hybrid CoT models**
> We focus on hybrid reasoning models because they generate substantially longer CoT traces, making efficiency gains more substantial than in chat models. Standard instruction-tuned LLMs that rely on prompted CoT produce much shorter chains, so the savings from not triggering CoT are far smaller. However, our method itself is model agnostic as it only requires a direct answer, a CoT answer, and a confidence score, so we do expect it to transfer to prompted CoT models.
>
> **Question 1:**
> Please refer to the shared response “Lack of or missing datapoints”.
>
> ---
>
> **References**
>
> [1] Sprague, Z. et al. *To CoT or not to CoT? Chain-of-thought helps mainly on math and symbolic reasoning.* 2024.

---

### Official Review · Reviewer_dKmw · 2025-11-02

**Soundness:** 3
**Presentation:** 4
**Contribution:** 2
**Rating:** 6
**Confidence:** 4

**Summary:**

This paper investigates whether confidence estimates from large language models (LLMs) can effectively determine when to employ Chain-of-Thought (CoT) reasoning. It systematically evaluates four training-free confidence methods for gating CoT, aiming to reduce unnecessary token consumption while maintaining task accuracy. The results demonstrate that these confidence measures can curtail redundant CoT tokens, yet their effectiveness varies significantly across different models and tasks. The study concludes that while confidence gating shows promise for efficient reasoning, current methods exhibit too much inconsistency for robust real-world deployment.

**Strengths:**

1. The calibration of verbalized confidence in LLMs is a critical research topic, and this study has significant practical implications.

2. The experimentation is comprehensive and in-depth, thoroughly demonstrating that the proposed methods can simultaneously reduce token usage while preserving accuracy.

3. The proposed confidence-gating pipeline is intuitive and elegant, offering a straightforward approach to effectively address the important problem of deciding when a model should engage in "fast" (direct) versus "slow" (CoT) reasoning.

**Weaknesses:**

1. While the paper introduces a promising research direction and evaluates several feasible methods, it lacks a unified or theoretically grounded framework, which limits its theoretical contribution.

2. In Figure 2, several confidence estimation methods underperform the random baseline. A theoretical analysis from the authors explaining why and under what conditions these methods fail would significantly strengthen the paper.

3. Some important related work for LLM confidence calibration is not discussed, e.g., [1,2].

[1] Xiong, M., Hu, Z., Lu, X., et al. Can LLMs Express Their Uncertainty? An Empirical Evaluation of Confidence Elicitation in LLMs. ICLR 2024.
[2] Zhou Z, Jin T, Shi J, et al. SteerConf: Steering LLMs for Confidence Elicitation. NeurIPS 2025.

**Questions:**

1. I am highly interested in whether this approach remains effective on state-of-the-art, closed-source models (e.g., future models like GPT-5 or Gemini 2.5 Pro) in terms of reducing token usage while maintaining accuracy. Experimental results on CoT gating based on verbalized confidence for these models would strengthen the work.

2. Could this methodology be extended to the ReAct framework? Tool-use paradigms also face a similar challenge, where simple tool calls may not require reasoning, whereas complex ones do. Experimenting on tool-using benchmarks, such as $\tau^2$-bench, could reveal further interesting phenomena.

---

> ### Author Response · Authors · 2025-11-18
>
> Thank you very much for your review. We provide our response below.
>
> **W1: Theoretical grounding**
> Our contribution is primarily empirical, aiming to provide a comprehensive evaluation framework and evidence base for confidence-gated CoT rather than a theoretical model. Our motivation is grounded in the literature showing that CoT is not uniformly helpful [6,7], which motivates studying how we can decide when CoT should be used at all. We would appreciate clarification on which aspects of the work they believe require theoretical grounding.
>
> **W2: Analysis of underperforming methods**
> We completely agree that this kind of analysis would strengthen the paper. **We will include this in the updated manuscript starting at the end of line 351:**
>
> > Interestingly, even though larger models are generally better, some confidence methods perform worse than random e.g. p(true) for gpt-oss and verbalised for Qwen3-8B. This is due to poor discriminative power indicated by low AUROC scores in these instances (below 0.5) which indicates worse than random performance. This is consistent with prior work that generally finds inconsistent results with verbalised confidence across models (Yang et al., Tian et al.).
>
> **W3: Related work**
> To better include this related work, **we update the related work section as follows:**
>
> > Other approaches seek to simply prompt the LLM to output a confidence score in its response (Tian et al., 2023; Xiong et al., 2024; Yang et al., 2025). Zhou et al. build on this to develop prompting strategies specifically designed to steer the model to produce better calibrated verbalised confidence scores.
>
> **Q1: Closed source models**
> We agree this is an interesting avenue. However, due to current budget constraints, we are currently unable to extend the experiment using closed-source models. Additionally, we wanted to focus on models for which all of our methods could be applied in both white-box and black-box settings. Even so, our framework can definitely be applied in this context.
>
> **Q2: Extension to REACT framework/tool-use**
> This is a valuable suggestion. Many tool-use tasks require multi-step action sequences, so obtaining a single “direct answer” before deciding on CoT is often not appropriate. A more suitable adaptation would be to apply confidence-based gating after each tool invocation to decide whether longer reasoning is needed, similar in spirit to approaches used in uncertainty-guided code generation [5]. We consider this an interesting direction, but it is outside the scope of our current study, and we leave it for future work.
>
> ---
>
> **References**
>
> [1] Xiong, M. et al. *Can LLMs Express Their Uncertainty? An Empirical Evaluation of Confidence Elicitation in LLMs.* ICLR, 2024.
> [2] Zhou, Z. et al. *SteerConf: Steering LLMs for Confidence Elicitation.* NeurIPS, 2025.
> [3] Tian, K. et al. *Just Ask for Calibration: Strategies for Eliciting Calibrated Confidence Scores from Language Models Fine-Tuned with Human Feedback.* EMNLP, 2023.
> [4] Yang, D. et al. *On Verbalized Confidence Scores for LLMs.* 2024.
> [5] Zhu, Y. et al. *Uncertainty-Guided Chain-of-Thought for Code Generation with LLMs.* 2025.
> [6] Sprague, Z. et al. *To CoT or not to CoT? Chain-of-thought helps mainly on math and symbolic reasoning.* 2024.
> [7] Yang Sui et al. *Stop Overthinking: A Survey on Efficient Reasoning for Large Language Models*, 2025.

---

### Author Response · Authors · 2025-11-18
**Shared Response**

**Lack of/missing datapoints**
We thank the reviewers for raising this. We do provide an explanation for this in the paper, however, it is not clearly linked to the fewer points on the graph. Starting on line 303, we provide the reason: *“Margin and perplexity achieve above-random performance only at low budgets, but quickly saturate: the distributions collapse to narrow ranges, limiting their separating power.”*

**We have updated the manuscript to include the following, which specifically references the graphs and why there are fewer points for these settings:**

> At higher budgets, margin and perplexity scores can no longer be meaningfully separated because many examples share the same value (namely 1.0 for both). As the threshold increases with the higher CoT-use budget, these identical scores cannot be distinguished, so beyond a certain point, all thresholds yield the same routing behaviour. This results in fewer distinct points on the trade-off plots, as seen in Figure 2 for Qwen-8B.

---

### Note · Authors · 2025-12-01

I have read and agree with the venue's withdrawal policy on behalf of myself and my co-authors.